# ThinkAct: Vision-Language-Action Reasoning via Reinforced Visual Latent Planning

**Chi-Pin Huang**[1,2]     **Yueh-Hua Wu**[1]     **Min-Hung Chen**[1]
**Yu-Chiang Frank Wang**[1,2]     **Fu-En Yang**[1]

[1] NVIDIA     [2] National Taiwan University
{chipinh, krisw, minhungc, frankwang, fredy}@nvidia.com

## Abstract

Vision-language-action (VLA) reasoning tasks require agents to interpret multi-modal instructions, perform long-horizon planning, and act adaptively in dynamic environments. Existing approaches typically train VLA models in an end-to-end fashion, directly mapping inputs to actions without explicit reasoning, which hinders their ability to plan over multiple steps or adapt to complex task variations. In this paper, we propose ThinkAct, a dual-system framework that bridges high-level reasoning with low-level action execution via reinforced visual latent planning. ThinkAct trains a multimodal LLM to generate embodied reasoning plans guided by reinforcing action-aligned visual rewards based on goal completion and trajectory consistency. These reasoning plans are compressed into a visual plan latent that conditions a downstream action model for robust action execution on target environments. Extensive experiments on embodied reasoning and robot manipulation benchmarks demonstrate that ThinkAct enables few-shot adaptation, long-horizon planning, and self-correction behaviors in complex embodied AI tasks. Project page: https://jasper0314-huang.github.io/thinkact-vla/

## 1   Introduction

Recent advances in multimodal large language models (MLLMs) [44, 25, 2, 41, 23, 1, 17, 8, 27, 56, 22, 6] have led to impressive progress on various tasks requiring the understanding of multimodal inputs, such as visual question answering and image/video captioning. However, while multimodal content can now be effectively perceived and interpreted, conducting multi-step planning for long-horizon user goals and then interacting with dynamic environments remains challenging for frontier MLLMs. Therefore, enabling the vision-language foundation models with action awareness and embodied reasoning capabilities unleashes a wide range of physical AI applications (e.g., robotics and AR assistance), and draws significant attention from both academics and industry.

To bridge action with vision-language modalities, several works [4, 16, 55, 3, 45] learn vision-language-action (VLA) models by initializing from pre-trained MLLMs and training on large-scale robotic demonstrations (e.g., Open X-Embodiment Dataset [33]). For example, OpenVLA [16] builds upon MLLMs with post-training on large-scale robot demonstrations, while TraceVLA [55] further applies visual traces prompting to enhance spatial understanding. Despite promising on short-horizon skills, the crucial capabilities to reason in diverse visual scenes and enable long-horizon planning remain limited due to the *end-to-end* fashion from visual and textual inputs to low-level actions.

To equip VLAs with the ability to solve complex embodied tasks, recent works [52, 10, 54, 40] have explored incorporating explicit chain-of-thought (CoT) prompting [47] as an intermediate step-by-step guidance. For instance, ECoT [52] and RAD [10] introduce data curation pipelines to generate intermediate steps and decomposed plans by prompting off-the-shelf MLLMs. Once the

39th Conference on Neural Information Processing Systems (NeurIPS 2025).

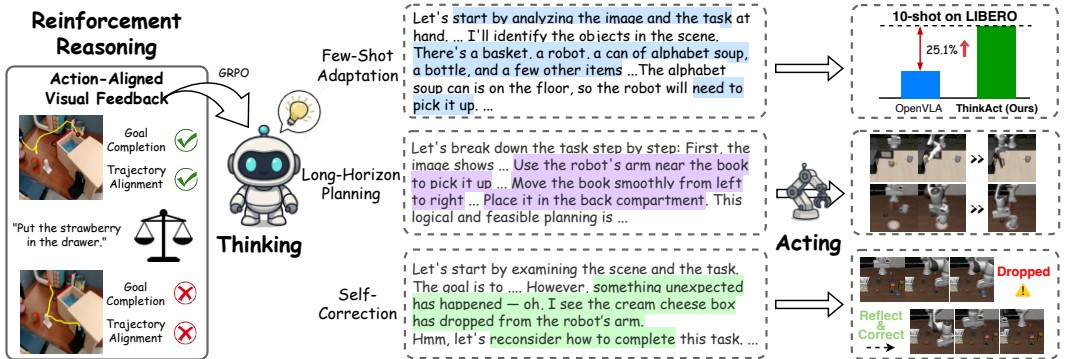

Figure 1: We introduce ThinkAct, a reasoning VLA framework capable of thinking before acting. Through reasoning reinforced by our *action-aligned visual feedback*, ThinkAct enables capabilities of few-shot adaptation, long-horizon planning, and self-correction in embodied tasks.

annotated CoT traces are obtained, VLAs are trained to predict intermediate steps via fully *supervised fine-tuning (SFT)*. However, due to the high cost of producing high-quality reasoning traces, the resulting models are prone to overfitting to specific visual scenes or reasoning patterns.

Recently, reinforcement learning (RL) [39, 14] has demonstrated significant potential to incentivize reasoning behaviors in LLMs by exploring the thinking trace that maximizes reward signals instead of solely relying on fully supervised CoT annotations. Inspired by this paradigm, several vision-language models [12, 31, 43] have applied RL-based reasoning to multimodal tasks. For example, Video-R1 [12] adopts R1-style RL optimization to induce the CoT traces by verifiable answer accuracy with format correctness. While this manner enables long-form reasoning without step-level supervision, the reliance on QA-style reward signals limits their ability to support long-horizon planning and makes it difficult to connect reasoning with real-world action execution.

In this paper, we propose *ThinkAct*, which aims to enable MLLMs with the capability to reason before acting in physical environments. To address vision-language-action reasoning tasks, ThinkAct adopts a dual-system architecture that connects structured reasoning with executable actions. Specifically, we incentivize MLLMs to perform long-horizon planning by advancing reinforcement learning with an action-aligned reward, derived from visual goal completion and trajectory distribution matching. Our ThinkAct leverages human and robot videos to elicit embodied reasoning that is grounded in visual observations. To bridge reasoning and execution, we compress intermediate reasoning steps into a compact latent trajectory that captures high-level intent and allows efficient adaptation of the downstream action network to new environments. By reinforcing structured reasoning and grounding it in real-world actions, ThinkAct tackles long-horizon manipulation tasks while unleashing few-shot action adaptation and self-correction behavior in physical AI scenarios, as shown in Fig. 1.

Our main contributions are summarized as follows:

- We propose *ThinkAct*, a dual-system framework that mutually enhances action execution and visual-grounded embodied reasoning connected by visual latent planning.
- We leverage the visual feedback of goal completion and trajectory alignment as action-aligned rewards to allow long-horizon reasoning grounded in the embodied scene.
- We advance visual latent planning to steer downstream action execution by providing reasoning-enhanced trajectory guidance across diverse environments.
- We demonstrate that our learned reasoning VLA enables capabilities of few-shot adaptation, long-horizon planning, and self-correction across diverse embodied manipulation tasks.

## 2 Related Works

### 2.1 Vision-Language-Action Models

Recent efforts [19, 50, 51, 30, 11] have adapted large language models (LLMs) and vision-language models (VLMs) for action-centric tasks by prompting or post-training on curated instruction-following

data. For example, RoboPrompt [50] is designed to prompt off-the-shelf LLMs to predict robot actions by constructing in-context demonstrations. RoboPoint [51] and LLARVA [30] leverage point and visual trajectory into textual prompts to augment LLMs with spatial-action understanding ability. AHA [11] enhances failure detection ability in robotic manipulation by formulating it as a free-form question-answering task, training on synthetic failure data generated by perturbing successful trajectories. Although effective in specific domains, these approaches depend on sophisticatedly curated data and struggle to generalize beyond their training distributions. To improve scalability, recent vision-language-action (VLA) models [16, 55, 42, 3, 21, 48] adopt large-scale robot datasets (e.g., Open X-Embodiment Dataset [33] or DROID [15]) to train models directly on diverse demonstrations. OpenVLA [16] learns from pre-trained VLMs with robot trajectories for generalist action execution, while TraceVLA [55] and HAMSTER [21] enhance spatial-action awareness by incorporating visual traces. However, these models predict actions directly from vision and language inputs, often bypassing structured planning or intermediate reasoning. As a result, their capability to handle complex instructions, long-horizon goals, or out-of-distribution scenarios remains limited.

## 2.2 Reasoning in Vision-Language-(Action) Models

Chain-of-thought (CoT) prompting [47, 46, 49] has significantly improved the multi-step reasoning ability of LLMs across math, coding, and question-answering tasks. Motivated by these advances, recent works extend reasoning capabilities to vision-language-action (VLA) models for embodied tasks. ECoT [52] synthesizes intermediate subgoals via prompting and applies supervised fine-tuning to teach VLAs to reason before acting. RAD [10] leverages action-free human videos to curate reasoning traces by prompting off-the-shelf LLMs and learn to map reasoning to real actions using robot data. On the other hand, CoT-VLA [54] replaces linguistic CoT with visual subgoal frames generated ahead of action prediction. However, they depend on either curated CoT supervision or task-specific video generation, limiting their scalability. Inspired by the recent success of RL-optimized reasoning models [39, 14], several approaches [12, 31, 43, 28] adopt GRPO [39] optimization to guide CoT generation in vision-language tasks using verifiable rewards. However, their QA-formatted rewards cannot fully support long-horizon planning or establish grounding between reasoning and action execution. To unify structured CoT reasoning with embodied decision-making, we introduce ThinkAct, which leverages action-aligned reinforcement learning and visual latent planning to connect embodied reasoning with real-world action in VLA tasks.

## 3 Method

### 3.1 Problem Formulation

We first define the setting and notations for vision-language-action (VLA) reasoning tasks. At each timestep $t$, the model receives a visual observation $o_t$ and a textual instruction $l$, with the goal of predicting an action $a_t$, which can be a textual command or a 7-DOF control vector $[\Delta_x, \Delta_\theta, \Delta_{\text{Grip}}]$ depending on the embodiment. To tackle this problem, we propose *ThinkAct*, a unified framework that aims to leverage an MLLM $\mathcal{F}_\theta$ to reason the high-level plans while connecting with an action model $\pi_\phi$ to infer executable actions. The MLLM $\mathcal{F}_\theta$ produces a visual plan latent $c_t$ based on $(o_t, l)$, capturing the high-level intent and planning context (Sec. 3.2). This reasoned plan $c_t$ then guides the downstream action module $\pi_\phi$ to sequentially predict $N$ executable actions $[a_t]_t^{t+N}$ tailored to the target environment (Sec. 3.3). By connecting abstract planning with low-level control, our ThinkAct enables long-horizon reasoning and improves action adaptation in dynamic embodied tasks.

### 3.2 Reinforced Visual Latent Planning for Embodied Reasoning

To enable embodied reasoning that generalizes across diverse environments, we aim to incentivize the reasoning capability of multimodal LLMs via reinforcement learning [39, 14]. A straightforward way is to have the MLLM reason before generating low-level actions, while using the resulting task success rate in target environments (e.g., LIBERO [24]) as the reward signal. However, this approach is restricted to specific simulators without proper guidance from visual scenes.

**Reward Shaping from Action-Aligned Visual Feedback** To tackle this challenge, we design a novel action-aligned visual feedback that captures long-horizon goals and encourages visual

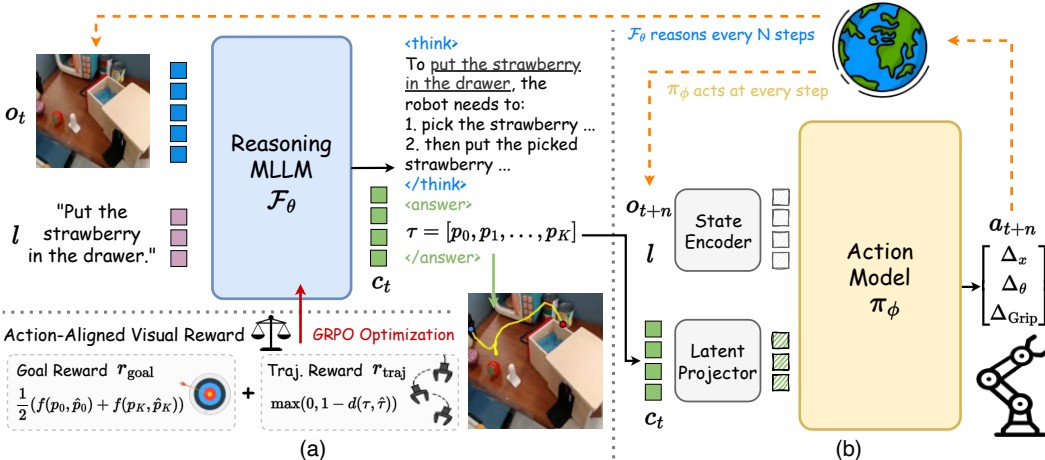

Figure 2: **Overview of our ThinkAct.** (a) Given observation $o_t$ and instruction $l$, ThinkAct advances *action-aligned* rewards derived from visual trajectory $\tau$ to incentivize embodied reasoning capability of Reasoning MLLM $\mathcal{F}_\theta$. (b) Conditioned on the visual plan latent $c_t$, the DiT-based Action Model $\pi_\phi$ learns to predict executable action while keeping $\mathcal{F}_\theta$ frozen. Note that, during inference, $\pi_\phi$ and $\mathcal{F}_\theta$ could operate asynchronously to enable slow thinking and fast control for VLA reasoning tasks.

grounding during planning. Specifically, inspired by recent works [48, 55], we are capable of representing high-level plans as spatial-temporal trajectories that capture the gripper end-effector over the visual scene, which serve as a visual-action guidance to steer the embodied reasoning.

As depicted in Fig. 2(a), given an observation $o_t$ at timestep $t$ and a task instruction $l$, the MLLM $\mathcal{F}_\theta$ autoregressively generates a sequence of latent embeddings for reasoning $v_t \in \mathbb{R}^{|v_t| \times d}$ and visual plan $c_t \in \mathbb{R}^{|c_t| \times d}$, where the former is decoded to reasoning steps while the latter would be inferred into a text string of 2D points $\tau = [p_k]_{k=1}^K$, with $p_k \in [0,1]^2$, and $p_1$ and $p_K$ denoting the *start* and *end* positions of the gripper. As a result, to encourage the model to anticipate visual goal completetion, we introduce the *goal reward* for comparing predicted start and end positions with corresponding points from trajectory obtained by off-the-shelf detector [30] $\hat{\tau} = [\hat{p}_k]_{k=1}^K$ as follows,

$$r_{\text{goal}} = \frac{1}{2} \left( f\left(p_1, \hat{p}_1\right) + f\left(p_K, \hat{p}_K\right) \right), \quad \text{where } f(p, p') = \max\left(0, 1 - \|p - p'\|_2^2\right). \tag{1}$$

To further enforce the MLLM predicted trajectory to properly correspond to physically plausible gripper motion, the *trajectory reward* is proposed to regularize the predicted $\tau$ to match the distribution of demonstrated trajectory $\hat{\tau}$. Thus, the trajectory reward $r_{\text{traj}}$ can be computed as follows,

$$r_{\text{traj}} = \max\left(0, 1 - d(\tau, \hat{\tau})\right). \tag{2}$$

Here, $d(\tau, \hat{\tau})$ denotes a metric measuring the distance between two trajectories, i.e., dynamic time warping (DTW) distance [37] in this work.

The overall reward is thus defined as the combination of our proposed action-aligned visual feedback and the format correctness score $r_{\text{format}}$ following existing reasoning works [31, 14]:

$$r = 0.9 r_{\text{visual}} + 0.1 r_{\text{format}}, \text{where } r_{\text{visual}} = \omega_{\text{goal}} r_{\text{goal}} + \omega_{\text{traj}} r_{\text{traj}}. \tag{3}$$

Here, $\omega_{\text{goal}} = \omega_{\text{traj}} = 0.5$ are the weighting coefficients for the goal and trajectory rewards.

**Reinforced Fine-Tuning for Eliciting Visual Latent Planning** To incentivize the embodied reasoning from the MLLM $\mathcal{F}_\theta$, we perform reinforced fin-tuning using Group Relative Policy Optimization (GRPO) [39]. Specifically, given an input $(o_t, l)$, GRPO first samples a group of $M$ distinct responses $\{z_1, z_2, \ldots, z_M\}$ from the original MLLM $\mathcal{F}_{\theta_{\text{old}}}$. Each response is evaluated using the reward function defined in Eq. 3 and resulting in a set of reward signals $\{r_1, r_2, ..., r_M\}$. Thus, we optimize $\mathcal{F}_\theta$ by maximizing the following objective:

$$\mathcal{J}_{\text{GRPO}}(\theta) = \frac{1}{M} \sum_{i=1}^{M} \left( \frac{\mathcal{F}_\theta(z_i|o_t, l)}{\mathcal{F}_{\theta_{\text{old}}}(z_i|o_t, l)} A_i - \beta D_{KL}(\mathcal{F}_\theta(z_i|o_t, l) \parallel \mathcal{F}_{\theta_{\text{old}}}(z_i|o_t, l)) \right), \tag{4}$$

$$\text{where} \quad A_i = \frac{r_i - \text{mean}(\{r_1, \ldots, r_M\})}{\text{std}(\{r_1, \ldots, r_M\})}.$$

Here, $A_i$ quantifies the relative quality of $i$-th response compared to other candidates in the sampled group. $D_{KL}(\cdot \parallel \cdot)$ is the KL divergence introduced with a weighting factor $\beta$ to regularize the model, preventing excessive deviation from the original model $\mathcal{F}_{\theta_{\text{old}}}$.

To further obtain general embodied knowledge, our ThinkAct is flexible to encapsulate the publicly available question-answering data to enhance capabilities such as robotic VQA [38] or failure detection [26] by formatting them into the QA-style accuracy reward. Specifically, the QA-style accuracy reward is computed by either answer accuracy for multiple-choice QA tasks or averaged ROUGE-1/2/L scores for open-ended QA tasks, as mentioned in Supplementary Sec. B.1.1. Once we obtain the QA reward $r_{\text{QA}}$, we use the same approach as in Eq. 3 that combines the QA-style reward with the format reward, and then optimize using GRPO. Specifically, for QA tasks, the total reward becomes: $r = 0.9 r_{\text{QA}} + 0.1 r_{\text{format}}$.

Once the reinforced fine-tuning is complete, we are able to produce long CoT steps, while abstracting the textual reasoning into a compact visual plan latent $c_t$, capturing long-horizon spatial-temporal planning intent.

### 3.3 Reasoning-Enhanced Action Adaptation

With the high-level embodied intent reasoned by the MLLM, our goal is to connect the inferred visual latent planning $c_t$ with the action model $\pi_\phi$ of the target environment in a think-before-acting manner, grounding embodied reasoning into the physical world with executable actions. Specifically, we build upon a Transformer-based action model $\pi_\phi$ (e.g., Diffusion Policy [9]), which predicts actions based on the current state composed of visual observations and language instructions. While $\pi_\phi$ can operate in the target environment using perception alone, we enhance its capability by conditioning it on the latent plan $c_t$, which encodes high-level embodied intent and planning context.

As depicted in Fig. 2(b), we incorporate $c_t$ using a latent projector to connect it to the input space of the action model, enabling the reasoning guidance to be effectively leveraged, which enhances its low-level action execution in the target environment. Thus, we solely update the state encoder, latent projector, and action model by imitation learning with annotated action demonstrations:

$$\mathcal{L}_{\text{IL}}(\phi) = \mathbb{E}_{(o_i, l, a_i)} \left[ \ell \left( \pi_\phi(c_t, o_i, l), a_i \right) \right]. \tag{5}$$

We note that, reasoning and action execution could be operated in an *asynchronous* manner, which means each latent plan $c_t$ corresponds to $N$ interactions with the environment (i.e., $i \in [t, t+N]$). This asynchronous design highlights a key advantage of our dual-system architecture, allowing the reasoning MLLM to perform slow thinking while the action model executes fast control.

### 3.4 Learning Strategy and Inference

Following [31], we adopt a multi-stage training strategy for our ThinkAct. Before RL, we initialize the two modules independently. The MLLM $\mathcal{F}_\theta$ is cold-started using supervised data (Sec. 4.1) to learn to interpret visual trajectories and produce reasoning and answers in the correct output format. On the other hand, the action model $\pi_\phi$ is pre-trained on the Open X-Embodiment (OXE) dataset [33], providing a strong foundation for low-level action execution. After SFT cold-start, our MLLM $\mathcal{F}_\theta$ is tuned with action-aligned rewards guiding the generation of effective latent plans. During reasoning-enhanced action adaptation, we freeze $\mathcal{F}_\theta$ while updating the action model $\pi_\phi$ with state encoder and latent projector on the target environment by conditioning on the latent visual plan $c_t$.

At inference time, given a visual observation $o_t$ and instruction $l$, ThinkAct produces a visual plan latent $c_t = \mathcal{F}_\theta(o_t, l)$, which conditions the action module $\pi_\phi$ to predict a sequence of executable actions tailored to the current environment.

## 4 Experiment

### 4.1 Experimental Setup

**Implementation Details** We initialize $\mathcal{F}_\theta$ with Qwen2.5-VL 7B [2]. The cold-start stage runs for 20K iterations with batch size 32 and learning rate $1e-5$ using DeepSpeed ZeRO-3. We then apply

Table 1: Quantitative comparisons of robot manipulation tasks on SimplerEnv [20] and LIBERO [24] benchmarks. **Bold** denotes the best result.

| Dataset | Split | Octo-Base [45] | RT1-X [5] | OpenVLA [16] | DiT-Policy [9] | TraceVLA [55] | CoT-VLA [54] | Magma [48] | ThinkAct (Ours) |
|---|---|---|---|---|---|---|---|---|---|
| Simpler-Google (Visual Matching) | Open/Close Drawer | 1.0 | 22.5 | 49.5 | 44.9 | 57.0 | – | 56.0 | 50.0 |
| | Move Near | 3.0 | 55.0 | 47.1 | 58.9 | 53.7 | – | 65.4 | 72.4 |
| | Pick Coke Can | 1.3 | 52.8 | 15.3 | 64.3 | 28.0 | – | 83.7 | 92.0 |
| | Overall | 1.8 | 43.4 | 37.3 | 56.0 | 46.2 | – | 68.4 | **71.5** |
| Simpler-Google (Variant Aggregation) | Open/Close Drawer | 22.0 | 56.0 | 22.5 | 35.5 | 31.0 | – | 53.4 | 47.6 |
| | Move Near | 4.2 | 34.2 | 54.0 | 52.8 | 56.4 | – | 65.7 | 63.8 |
| | Pick Coke Can | 17.0 | 54.0 | 52.8 | 56.4 | 60.0 | – | 68.8 | 84.0 |
| | Overall | 14.4 | 48.1 | 43.1 | 48.2 | 49.1 | – | 62.6 | **65.1** |
| Simpler-Bridge (Visual Matching) | Put Carrot on Plate | 8.3 | 4.2 | 4.2 | 29.4 | – | – | 31.0 | 37.5 |
| | Stack Blocks | 0.0 | 0.0 | 0.0 | 0.0 | – | – | 12.7 | 8.7 |
| | Put Spoon on Towel | 12.5 | 0.0 | 8.3 | 34.5 | – | – | 37.5 | 58.3 |
| | Put Eggplant in Basket | 43.1 | 0.0 | 45.8 | 65.5 | – | – | 60.5 | 70.8 |
| | Overall | 16.0 | 1.1 | 14.6 | 32.4 | – | – | 35.4 | **43.8** |
| LIBERO | Spatial | 78.9 | – | 84.7 | 82.6 | 84.6 | 87.5 | – | 88.3 |
| | Object | 85.7 | – | 88.4 | 84.7 | 85.2 | 91.6 | – | 91.4 |
| | Goal | 84.6 | – | 79.2 | 82.1 | 75.1 | 87.6 | – | 87.1 |
| | Long | 51.1 | – | 53.7 | 57.6 | 54.1 | 69.0 | – | 70.9 |
| | Overall | 75.1 | – | 76.5 | 76.8 | 74.8 | 83.9 | – | **84.4** |

GRPO [39] for 6K iterations, using batch size 64, learning rate $1e-6$, and rollout size 5. The action model $\pi_\phi$ is a DiT-based policy [9] with 432M parameters, pre-trained using the OXE dataset [33], where the state encoder is composed of a DINOv2 image encoder [32] and a CLIP text encoder [36] that jointly encode the current state inputs into 1024-dim embeddings. For reasoning-enhanced action adaptation, we connect the visual plan $c_t$ via a Q-Former [18] as the latent projector with 32 queries and fine-tune on 100K data randomly sampled from the OXE dataset for 120K iterations using batch size 256 and learning rate $2e-5$. LIBERO [24] tasks are further fine-tuned for 75K iterations with batch size 128. All experiments are conducted on 16 NVIDIA A100 GPUs with 80 GB memory.

**Training Datasets and Evaluation Benchmarks**  For SFT cold-start, we fine-tune the MLLM using trajectories from the subset of OXE, and QA tasks from RoboVQA [38], EgoPlan-IT [7], and Video-R1-CoT [12]. During RL training, we incorporate trajectories from the OXE subset and human videos from Something-Something v2 [13]. To enhance general reasoning capability, we include embodied QA datasets such as EgoPlan-IT/Val [7], RoboVQA [38], and the Reflect dataset [26], as well as a general video instruction dataset, i.e., LLaVA-Video-178K [53].

We evaluate ThinkAct on two robot manipulation and three embodied reasoning benchmarks. For manipulation tasks, SimplerEnv [20] containing diverse scenes and LIBERO [24] with long-horizon tasks are evaluated using task success rate. For reasoning benchmarks, EgoPlan-Bench2 [35] uses accuracy on multiple-choice questions, while RoboVQA [38] and OpenEQA [29] are free-form QA tasks evaluated using BLEU score [34] and LLM-based scoring, respectively, following their original protocols. Further details of our experimental setup are provided in the supplementary material.

## 4.2 Quantitative Evaluation

**Robot Manipulation**  To assess the effectiveness of ThinkAct on robot manipulation task, we evaluate on SimplerEnv [20] and LIBERO [24]. SimplerEnv [20] includes Google-VM (Visual Matching), Google-VA (Variant Aggregation), and Bridge-VM setups, introducing variations in color, material, lighting, and camera pose to evaluate model robustness. For the LIBERO [24] benchmark, following prior works [16, 54], we evaluate on the LIBERO-Spatial, LIBERO-Object, LIBERO-Goal, and LIBERO-Long subtasks to test model generalization across spatial layouts, object variations, goal diversity, and long-horizon planning.

As shown in Tab. 1, on the SimplerEnv, incorporating our reasoning-guided visual plan latents allows ThinkAct to outperform our baseline action model, DiT-Policy, by 15.5%, 16.9%, and 11.4% on Google-VM, Google-VA, and Bridge-VM, respectively, achieving the highest overall scores of 71.5%, 65.1%, and 43.8% against all methods. On the LIBERO benchmark, ThinkAct achieves the best overall success rate of 84.4%, outperforming DiT-Policy and recent state-of-the-art CoT-VLA [54], verifying the effectiveness on diverse manipulation settings.

Table 2: Quantitative comparisons of embodied reasoning tasks on EgoPlan-Bench2, RoboVQA, and OpenEQA benchmarks. Note that, Qwen2.5-VL* indicates fine-tuning the original Qwen2.5-VL using EgoPlan-IT [7] and RoboVQA [38] datasets. **Bold** denotes the best result.

| Dataset | Split / Metric | GPT-4V [1] | LLaVA-Video [17] | InternVL2.5 [8] | InternVL3 [56] | NVILA [27] | Qwen2.5-VL [2] | Qwen2.5-VL* [2] | Magma [48] | ThinkAct (Ours) |
|---|---|---|---|---|---|---|---|---|---|---|
| EgoPlan-Bench2 | Daily life | 36.7 | 38.0 | 36.2 | 38.5 | 35.8 | 31.4 | 47.9 | 32.1 | 50.1 |
| | Work | 27.7 | 29.9 | 28.7 | 32.9 | 28.7 | 26.7 | 46.3 | 25.7 | 49.8 |
| | Recreation | 33.9 | 39.0 | 34.4 | 36.1 | 37.2 | 29.5 | 44.3 | 34.4 | 44.8 |
| | Hobbies | 32.5 | 37.4 | 35.4 | 37.2 | 35.4 | 28.6 | 44.2 | 29.3 | 45.2 |
| | Overall | 32.6 | 35.5 | 33.5 | 36.2 | 33.7 | 29.1 | 45.7 | 29.8 | **48.2** |
| RoboVQA | BLEU-1 | 32.2 | 35.4 | 40.5 | 44.3 | 42.7 | 47.8 | 65.3 | 38.6 | 69.1 |
| | BLEU-2 | 26.5 | 32.1 | 33.3 | 36.5 | 39.7 | 41.2 | 57.3 | 31.5 | 61.8 |
| | BLEU-3 | 24.7 | 30.0 | 29.6 | 31.6 | 37.6 | 36.2 | 52.2 | 28.1 | 56.0 |
| | BLEU-4 | 23.9 | 29.0 | 27.5 | 28.9 | 36.1 | 33.7 | 48.0 | 26.7 | 52.4 |
| | Overall | 26.8 | 31.6 | 32.7 | 35.3 | 39.0 | 39.7 | 55.7 | 31.2 | **59.8** |
| OpenEQA | Obj. State | 63.2 | 69.1 | 70.2 | 68.9 | 66.1 | 63.2 | 62.4 | 59.9 | 70.0 |
| | Obj. Recog. | 43.4 | 42.6 | 47.2 | 49.1 | 49.5 | 46.2 | 45.2 | 43.8 | 47.2 |
| | Func. Reason. | 57.4 | 50.3 | 56.2 | 54.6 | 51.0 | 51.2 | 52.3 | 50.0 | 53.2 |
| | Spatial | 33.6 | 46.2 | 44.1 | 43.3 | 43.1 | 41.2 | 42.8 | 39.3 | 47.6 |
| | Attri. Recog. | 57.2 | 64.1 | 64.9 | 74.4 | 69.3 | 63.0 | 65.0 | 58.3 | 71.1 |
| | World Know. | 50.7 | 60.5 | 56.5 | 53.1 | 59.4 | 54.3 | 54.2 | 53.3 | 58.6 |
| | Obj. Loc. | 42.0 | 38.2 | 41.9 | 45.0 | 39.9 | 36.5 | 41.9 | 38.9 | 45.9 |
| | Overall | 49.6 | 53.0 | 54.4 | 55.5 | 54.0 | 50.8 | 52.0 | 49.1 | **56.2** |

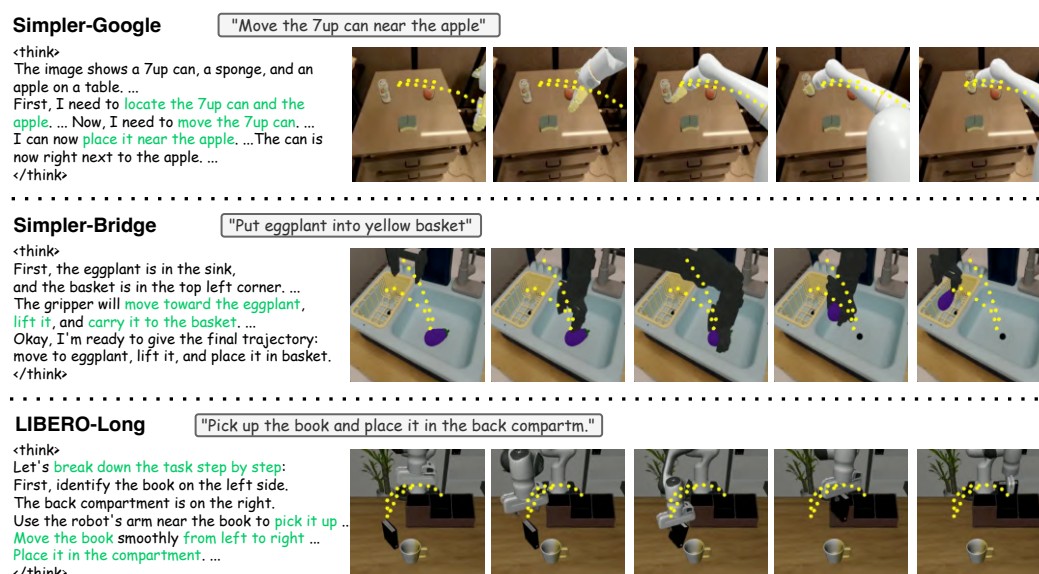

Figure 3: Qualitative results of intermediate reasoning steps and visualized trajectory for robot manipulation tasks on SimplerEnv and LIBERO benchmarks.

**Embodied Reasoning**    In Tab. 2, we assess the reasoning capability of ThinkAct in embodied scenarios on three benchmarks: EgoPlan-Bench2 [35], RoboVQA [38], and OpenEQA [29]. EgoPlan-Bench2 [35] measures multi-step planning in egocentric daily-life scenarios, while RoboVQA [38] focuses on long-horizon reasoning in robotic manipulation. ThinkAct outperforms the second-best method by 2.5% and 4.1 BLEU score on these two benchmarks, demonstrating its strength in long-horizon and multi-step planning. Separately, OpenEQA [29] measures zero-shot embodied understanding across diverse environments. The enhanced reasoning ability of ThinkAct enables better generalization and scene comprehension, resulting in strong performance on this benchmark.

## 4.3    Qualitative Results

In Fig. 3, we qualitatively showcase the reasoning process and execution scenes of two manipulation examples from the Simpler-Bridge [20] and LIBERO-Long [24] tasks. In the LIBERO-Long task "Pick up the book and place it in the back compartment," ThinkAct decomposes the instruction into sub-tasks: (1) pick up the book, (2) move from left to right, and (3) place it in the compart-

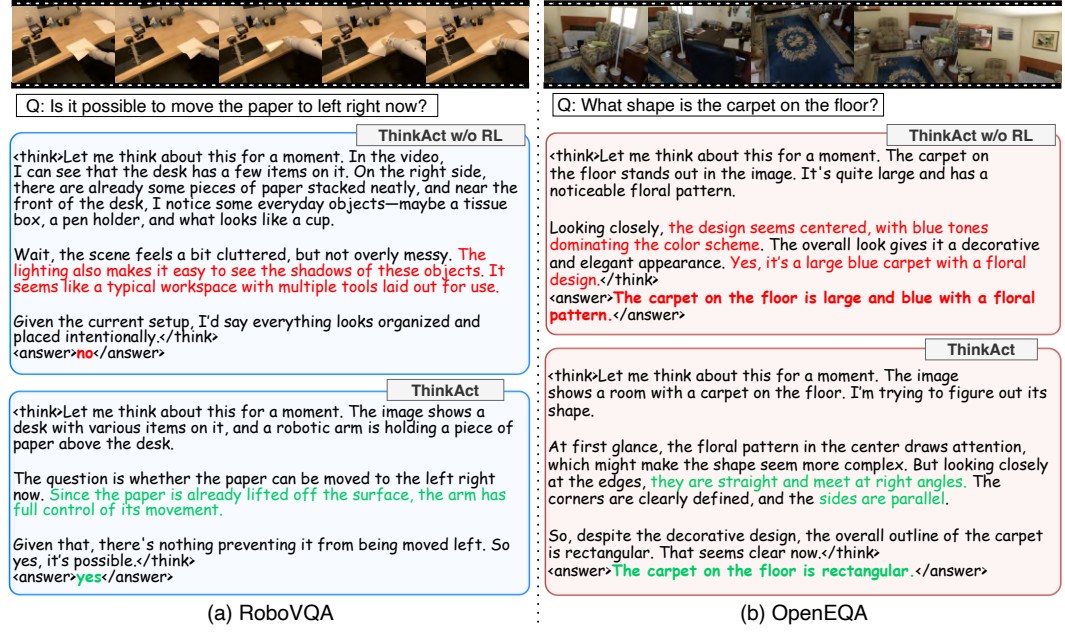

Figure 4: Qualitative comparison of reasoning process and the derived answer for our ThinkAct with and without RL for embodied reasoning tasks on RoboVQA and OpenEQA benchmarks. Red denotes incorrect reasoning and answers, while green indicates correct ones.

ment, demonstrating its *long-horizon* planning capability. We also visualize the planned trajectory, confirming that the gripper closely follows the reasoning-guided plan during execution.

To better illustrate the impact of RL on the reasoning process, Fig. 4 compares ThinkAct before and after RL fine-tuning on embodied reasoning tasks. As we can observe in Fig. 4(a), using a RoboVQA [38] example, the SFT cold-start model focuses only on the current state and fails to reason over future steps, while the RL-tuned model successfully infers the correct answer. Also, as demonstrated in Fig. 4(b), from OpenEQA [29], the cold-start model misinterprets the question, whereas the RL-tuned version demonstrates improved question and environment understanding. More qualitative comparisons and demo videos are provided in the supplementary material.

## 4.4 Ablation Study

In Tab. 3, we ablate the proposed goal reward $r_{\text{goal}}$ and trajectory reward $r_{\text{traj}}$ to analyze their individual contributions to reasoning and planning. We start from the full version of ThinkAct, which achieves the best performance across all benchmarks. Removing the trajectory reward leads to a noticeable drop, indicating that $r_{\text{traj}}$ is essential for learning coherent and structured planning behaviors. Without the goal reward, performance also declines, suggesting that $r_{\text{goal}}$ plays a key role in incentivizing long-horizon reasoning. When both $r_{\text{traj}}$ and $r_{\text{goal}}$ are removed, leaving only QA-style reward from QA datasets, the model shows only marginal improvements over the SFT baseline, confirming that action-aligned visual feedback is critical for effective multi-step planning in embodied settings. Finally, the SFT cold-start model without RL yields the lowest scores, verifying the effectiveness of our RL fine-tuning for eliciting the reasoning capability in MLLMs. More ablation studies (e.g., the number of interactions per reasoning step $N$) are provided in the supplementary material.

## 4.5 Analysis of ThinkAct

In this section, we analyze the capabilities of ThinkAct in enhancing robotic manipulation by embodied reasoning. We focus on two key aspects: (1) how reasoning facilitates effective few-shot adaptation to new tasks and environments, and (2) how it enables the robot to detect failures and perform self-correction during task execution. Through both quantitative experiments and qualitative

Table 3: Quantitative ablation study for our proposed RL rewards in ThinkAct on SimplerEnv, EgoPlan-Bench2, and RoboVQA benchmarks.

| Method | SimplerEnv | EgoPlan | RoboVQA |
|---|---|---|---|
| **ThinkAct (Ours)** | **60.1** | **48.2** | **59.8** |
| Ours w/o $r_{\text{traj}}$ | 59.2 | 47.9 | 58.5 |
| Ours w/o $r_{\text{goal}}$ | 59.1 | 47.6 | 58.9 |
| Ours w/o $r_{\text{traj}}, r_{\text{goal}}$ | 56.9 | 47.2 | 58.3 |
| SFT cold-start | 56.4 | 46.4 | 57.9 |

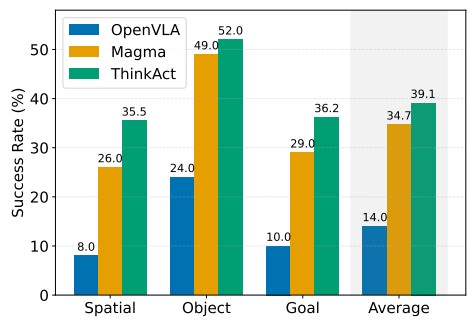

Figure 5: Few-shot adaptation results on LIBERO. We use 10 demonstrations per task for fine-tuning.

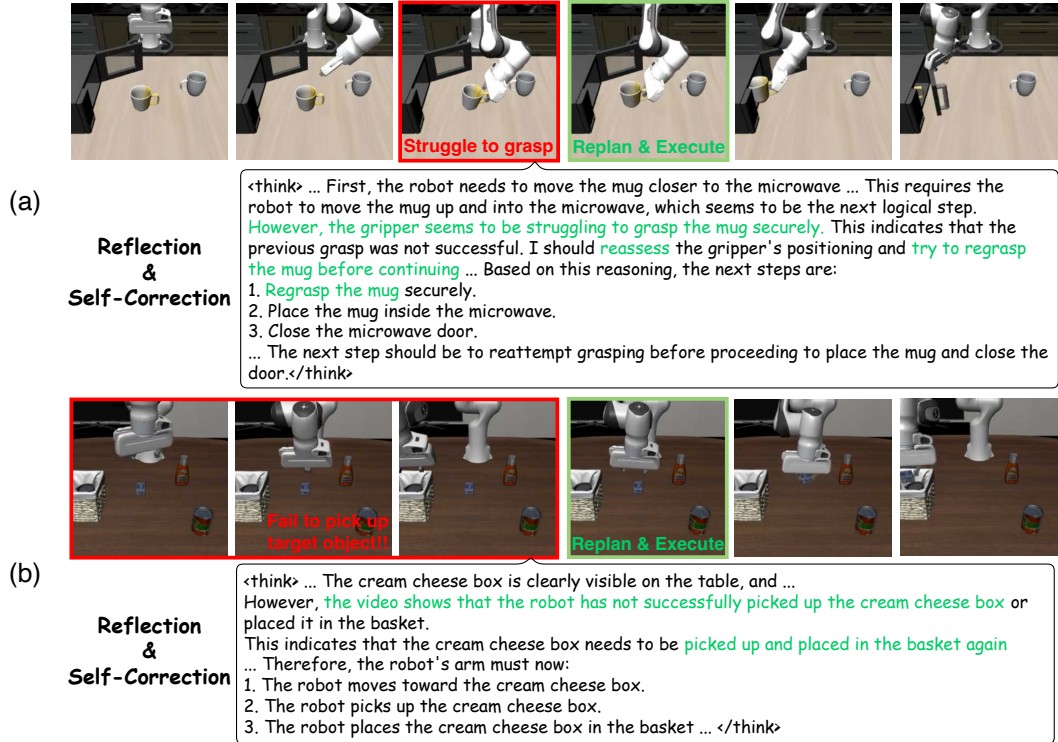

Figure 6: Demonstration of self-reflection and correction capability of ThinkAct. The reasoning MLLM identifies the failure and generates a revised plan that recovers from execution error.

examples, we demonstrate the unique advantages of leveraging a reasoning MLLM to tackle embodied action tasks. We further provide the analysis of MLLM backbones in the supplementary material.

**Reasoning Enhance Few-Shot Adaptation** As we can observe in Fig. 3 and Fig. 4, ThinkAct is capable of describing the environment and decomposing task instructions into meaningful sub-goals. To validate whether such reasoning improves the action model's adaptability, we conduct a few-shot adaptation experiment on LIBERO benchmark [24]. Specifically, we use LIBERO-Spatial and LIBERO-Object to evaluate adaptation to *unseen environments*, and LIBERO-Goal to test adaptation to *new skills*. We fine-tune the action model on just 10 demonstrations per task and evaluate performance over 100 trials. As shown in Fig. 5, ThinkAct consistently outperforms state-of-the-art methods, achieving the highest success rates across all tasks. Notably, it surpasses Magma [48] by 7.3% on LIBERO-Goal and by 9.5% on LIBERO-Spatial, demonstrating the effectiveness of reasoning capability for few-shot generalization in both novel skills and environments.

**Reasoning Elicit Reflection and Self-Correction** Failure detection and self-correction are critical for robust robot manipulation [26]. To evaluate whether ThinkAct can reason about and recover from execution errors, we enable the reasoning MLLM to observe more contextual information during execution by extending its input from a single image $o_t$ to a short video segment $o_{t-N:t}$. This temporal context allows ThinkAct to detect failures, reconsider the situation, and replan accordingly. For example, as shown in Fig. 6(a), the robot fails to grasp a mug. The reasoning MLLM identifies the issue, noting that the gripper is struggling, and suggests adjusting its position to reattempt the grasp. In Fig. 6(b), the robot attempts to move an object to a basket, but fails to pick it up in the first place. The MLLM detects the failure and replans the pickup, leading to successful completion. These cases highlight ThinkAct's ability to detect and recover from execution errors through reasoning.

**Inference Speed** We compare the inference speed of ThinkAct with the end-to-end OpenVLA [16] on LIBERO [24] tasks using an A100 GPU. On average, ThinkAct takes 17% longer execution time than OpenVLA, primarily due to the autoregressive reasoning process. We note that while the inference time slightly increases, our embodied reasoning, as a test-time scaling paradigm, significantly boosts downstream task performance. That is, ThinkAct outperforms OpenVLA on all four LIBERO task categories, achieving success rate improvements of 2.8% on spatial, 3.2% on object, 8.4% on goal, and 15.3% on long-horizon tasks. These results show that the reasoning overhead is justified by significant performance gains, highlighting the effectiveness of embodied reasoning for robot manipulation.

## 5   Conclusion

We presented *ThinkAct*, a framework that reinforces visual latent planning for vision-language-action reasoning tasks. By combining action-aligned reinforcement learning with reasoning-enhanced action adaptation, ThinkAct enables embodied agents to think before acting and execute robust actions in dynamic environments. Through extensive experiments across embodied reasoning and robot manipulation benchmarks, we demonstrated strong long-horizon planning, few-shot adaptation, and emergent behaviors such as failure detection and self-correction, providing a scalable path toward more deliberative and adaptable embodied AI systems.

**Limitations** Since ThinkAct builds on pretrained multimodal LLMs, it inevitably inherits their limitations, particularly hallucinations in visual or spatial reasoning. This can lead to generated plans that reference incorrect object attributes or spatial relationships, affecting downstream execution. While our latent planning and action grounding mitigate this to some extent, future work on grounding-aware training or hallucination suppression in MLLMs may further improve robustness and reliability in real-world deployment. In addition, while we only include 2D traces to calculate the reward signals, our reward framework can be readily extended to incorporate contact-rich signals into the total reward function (Eq. 3). The proposed action-aligned visual reward allows extension with additional reward components that capture contact-rich dynamics. We will leave them for future research.

**Broader Impacts** Our work aims to enhance the reasoning capabilities of embodied agents, which could support real-world applications such as assistive robotics, home automation, and industrial systems. In particular, models like ThinkAct may help robots better interpret vague instructions and execute multi-step plans in dynamic environments. However, increased autonomy and reasoning ability in embodied systems also raise potential concerns. Misinterpretation of ambiguous commands, reliance on hallucinated visual reasoning, or overconfidence in CoT outputs could result in unintended behaviors, especially in safety-critical settings. Hence, future research on safeguards or alignment with human intent could further help mitigate these risks.

**Acknowledgment** This work is supported in part by the National Science and Technology Council via grant NSTC 113-2634-F-002-005, NSTC 114-2221-E-002-056-MY2 and NSTC 114-2640-E-002-006, and the financial supports from the Featured Area Research Center Program within the framework of the Higher Education Sprout Project by the Ministry of Education (114L900902). We also thank the National Center for High-performance Computing (NCHC) for providing computational and storage resources.

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
