# OpenReview forum: "ThinkAct: Vision-Language-Action Reasoning via Reinforced Visual Latent Planning"
_NeurIPS.cc/2025/Conference — NeurIPS 2025 poster_

### Official Review · Reviewer_nnRq · 2025-06-22

**Clarity:** 4
**Significance:** 3
**Originality:** 2
**Rating:** 5
**Confidence:** 4

**Summary:**

The paper introduces a new VLA model called "ThinkAct", in which a multimodal LLM generates embodied reasoning plans expressed as language and visual latents, and a subsequent action model generates low level actions. Differently from other work, which just made use of Chain of Thought to boost a similar type of VLA, ThinkAct uses Reinforcement Learning (specifically GRPO) to optimize the first planning step.

The approach is evaluated on simulated manipulation benchmarks (SimplerEnv and LIBERO), embodied reasoning benchmarks, and is also investigated qualitatively. Overall, the approach improves performance and induces better reasoning traces over embodied actions.

**Questions:**

* Was the "SFT cold-start" ablation benchmarked in a more granular fashion for manipulation and embodied reasoning? How does it fare against the other benchmarked methods?

* Is it possible to see a benchmark of ThinkAct against "SFT cold-start" with comparable compute budgets (running the SFT model training for longer)?

**Ethical Concerns:**

["NO or VERY MINOR ethics concerns only"]

**Final Justification:**

The authors have addressed my comments and I thus update my score to a 5. There are still some weaknesses overall, plus the performance gains remain marginal, but the overall introduction of RL for reasoning in robotics is interesting and worthwile, so I recommend acceptance.

**Limitations:**

Limitations are discussed in the Appendix, not the main paper.

**Quality:**

3

**Strengths And Weaknesses:**

**Strengths**

The paper proposes an interesting architecture and training method for VLAs, applying the recent advances in RL-based reasoning models to robotics. The overall architecture is in line with the state of the art (such as in using a two-stage approach to separate planning and action, and in using a DiT policy for action generation), making the presented approach a meaningful investigation of the impact of RL-based planning when compared to the state of the art.

The choice of visual plan latents is interesting and might be the key behind improved sample efficiency of the method compared to other baselines.

**Weaknesses**

The main criticism of the work is that it appears that the impact of the RL finetuning technique used for enhancing planning is not completely disentangled from the overall "visual plan latent"-based proposed architecture for the VLA itself. When comparing the ThinkAct approach against others on manipulation tasks (Table 1), it seems to often severely improve over alternatives. However, in the ablation test (Table 3), it appears that the introduction of the RL step only adds a few percentage points of performance compared to "SFT cold-start", which consists of the proposed architecture, without the RL fine-tuning part.

It seems that the proposed architecture might be overall contributing to much of the improvement in performance: it would have been interesting to see "SFT cold-start" benchmarked with more details in the main experiment tables. In addition: in the ablation, ThinkAct and "SFT cold-start" are also not strictly comparable in compute budget. It would have been interesting to compare ThinkAct against a version of "SFT cold-start" that was trained for longer in order to match the total ThinkAct compute budget.

---

> ### Author Rebuttal · Authors · 2025-07-31
>
> ## **Response to Reviewer nnRq**
>
> **(W1) While ThinkAct shows significant improvements against others in Table 1, the ablation study (Table 3) shows that RL fine-tuning only adds a few percentage points compared to "SFT cold-start." It seems that the major performance gains come from the architectural design rather than the proposed RL method.**
>
> We appreciate the reviewer raising this important concern about disentangling the contributions of our architecture versus the RL fine-tuning technique. We would like to clarify that, following previous works [13,11], our reasoning framework includes both SFT cold-start and RL stages, where the SFT cold-start already incorporates reasoning capabilities with CoT data (i.e., Video-R1-CoT), as mentioned in Sec. B.2.2. Therefore, the comparison in Table 3 reflects the improvement from RL over an already reasoning-capable baseline, not the overall impact of reasoning itself.
>
> To properly disentangle these contributions, we conducted experiments with a non-reasoning baseline (Qwen2.5-VL\*\*) that uses the same "visual plan latent"-based architecture but is trained without any CoT data, as shown in the table below:
>
> | Method                   | Simpler-Google (VM) | Simpler-Google (VA) | Simpler-Bridge (VM) | LIBERO | Average |
> |---------------------------|------------------------------|-----------------------------|-----------------------------|-------------|-------------|
> | Qwen2.5-VL\*\*      | 65.3                             | 55.4                           | 37.7                          | 75.2        | 58.4 |
> | **ThinkAct**                 | **71.5**                             | **65.1**                           | **43.8**                          | **84.4**       | **66.2** |
> | $\Delta$                      | +6.2                            | +9.7                          | +6.1                          | +9.2        | +7.8
>
> Additionally, we evaluated our ThinkAct on the RoboFAC [A] benchmark that specifically requires reasoning about failure detection and correction, further demonstrating that our RL reasoning significantly improves upon both the SFT cold-start and non-reasoning baselines. This confirms that both the reasoning capabilities and RL fine-tuning are essential components, with the qualitative results in Figure 4 demonstrating our action-aligned visual rewards result in visually grounded reasoning thoughts.
>
> |Method  | Failure locating | Task identification | Failure detection | Task planning | Failure identification | Failure explanation | High-level correction | Low-level correction | Average |
> |-------------------|------------------|----------------------|-------------------|----------------|-------------------------|----------------------|------------------------|-----------------------|---------|
> | SFT cold-start        | 80.6             | 43.3                 | 79.3              | 38.7           | 35.6                    | 43.8                 | 46.4                   | 25.6                  | 49.2    |
> | **ThinkAct**      | **83.4**         | **53.9**             | **85.3**          | **48.7**       | **39.1**                | **51.3**             | **48.1**               | **28.3**              | **54.8** |
>
> [A] RoboFAC: A Comprehensive Framework for Robotic Failure Analysis and Correction. arXiv 2025.
>
>
> **(W2, Q1) It seems that the proposed architecture might be overall contributing to much of the improvement in performance: it would have been interesting to see "SFT cold-start" benchmarked with more details in the main experiment tables.**
>
> We appreciate the reviewer's suggestion for more detailed benchmarking of the SFT cold-start baseline. We conducted granular evaluations of our SFT cold-start across both manipulation and embodied reasoning benchmarks, comparing it against other methods as shown in the table below:
>
>
> | Method            | Simpler-Google (VM) | Simpler-Google (VA) | Simpler-Bridge (VM) | LIBERO |
> |------------------|----------------------|-----------------------|-----------------------|--------|
> | OpenVLA [15]          | 37.3                 | 43.1                  | 14.6                  | 76.5   |
> | CoT-VLA [49]          | -                    | -                     | -                     | 83.9   |
> | Magma [44]            | 68.4                 | 62.6                  | 35.4                  | -      |
> | SFT cold-start   | 67.4                 | 61.7                  | 40.2                  | 79.1   |
> | **ThinkAct**     | **71.5**             | **65.1**              | **43.8**              | **84.4** |
>
> | Method            | EgoPlan-Bench2 | RoboVQA | OpenEQA |
> |-------------------|----------------|---------|---------|
> | GPT-4V [1]            | 32.6           | 26.8    | 49.6    |
> | Qwen2.5-VL [2]        | 29.1           | 39.7    | 50.8    |
> | Magma [44]             | 29.8           | 31.2    | 49.1    |
> | SFT cold-start    | 46.4           | 57.9    | 53.3    |
> | **ThinkAct**   | **48.2**           | **59.8**    | **56.2**    |
>
> The results demonstrate that our SFT cold-start already achieves competitive performance on manipulation tasks and strong results on embodied reasoning benchmarks (outperforming SoTAs like [1,2,44]). However, the full ThinkAct with RL fine-tuning consistently outperforms the SFT cold-start across all tasks, particularly excelling in complex scenarios requiring visual grounding and long-horizon planning.
>
> This granular analysis highlights the crucial importance of our action-aligned visual rewards and RL fine-tuning for achieving optimal performance in embodied AI tasks, demonstrating that both architectural design and reward-guided reasoning optimization are essential components of our ThinkAct.
>
>
> **(W3, Q2) Is it possible to see a benchmark of ThinkAct against "SFT cold-start" with comparable compute budgets (running the SFT model training for longer)?**
>
> We conducted additional experiments where we extended the SFT cold-start training from 20K to 26K iterations (matching the total compute budget of ThinkAct's 20K SFT + 6K RL iterations) as shown in the table below:
>
> | Method                  | Total Iter. | SimplerEnv | LIBERO | EgoPlan-Bench2 | RoboVQA | OpenEQA |
> |-------------------------|-------------|------------|--------|----------------|---------|---------|
> | SFT cold-start          | 20K         | 56.4       | 79.1   | 46.4           | 57.9    | 53.3    |
> | SFT cold start$^{†}$    | 26K         | 55.9       | 79.3   | 46.3           | 58.2    | 52.0    |
> | **ThinkAct**            | 26K         | **60.1**   | **84.4** | **48.2**     | **59.8** | **56.2** |
>
> However, we observed no significant improvement in performance across benchmarks when simply extending SFT training duration. This is because for robot manipulation and embodied reasoning tasks, it is crucial to ground the reasoning into the visual scenes. Simply increasing supervised training cannot achieve this grounding effectively, which is why our proposed action-aligned visual rewards are crucial for connecting reasoning with executable actions in visual environments.

---

> > ### Comment · Reviewer_nnRq · 2025-08-05
> >
> > Thank you for your response and updated experiments, I have updated my review accordingly.

---

> ### Author Response · Authors · 2025-08-06
>
> Thank you for your valuable review and for taking the time to read our responses.
>
> We appreciate that our rebuttal addressed your concerns and will incorporate the clarifications and updated experiments into our revised version.
>
> Thank you again for your constructive feedback.

---

### Official Review · Reviewer_h8uF · 2025-06-27

**Clarity:** 3
**Significance:** 2
**Originality:** 2
**Rating:** 3
**Confidence:** 4

**Summary:**

The paper proposes ThinkAct, which try to bridge high-level reasoning with low-level action execution via reinforced visual latent planning. The method contains a SFT part, a RL tuning part and a final fine-tuning part. The author evaluate the results on several simulators and QA tasks to show the efficiency of the method.

**Questions:**

1. More explanation about the intuition on the design of the goal reward would be helpful. For goal completion, why would the goal reward contains a comparison on the start point $f(p_1, \hat{p}_1)$?

2. In L146-150, the authors propose to use QA-style accuracy reward. Can you provide more details about this reward?

3. For OXE dataset used in the cold-start phase, what are the latent embeddings $v$ for OXE dataset?

4. For the experiment settings, the comparison needs to be conduct in a fair setting. The authors propose to use a subset of OXE during SFT cold-start, RL training and finetuning (100K samples). What is the detailed mixture of this subset? Are these three stages using the same data? For baselines like OpenVLA, Magma, etc., they seem to use a mixed data that is of various source and evaluate in a zero-shot fashion. If these baselines are only pretrained on a wider range of OXE (for instance) data , the performance will be worse naturally. To make a fair comparison, the author should finetune the baselines using the same subset which seems to contain more test relevant data.

5. The main motivation of the paper is to build a system that can bridge high-level reasoning and low-level action. However, the evaluation tasks (Sipmler and LIBERO) mostly contain simple tasks like pick and place that does not require high level planning. This may hinder the proof of the final motivation.

6. Misc.: It would be helpful to include the definition of $r_{format}$ (eq.3) to make the paper self-contained.

**Ethical Concerns:**

["NO or VERY MINOR ethics concerns only"]

**Final Justification:**

thank the authors for the response! I think the authors find an interesting way to combine high level reasoning and robot execution and the combination is not naive. However, my major concerns still hold:

(1) the major claim is to "bridge high-level reasoning with low-level action execution via reinforced visual latent planning" However, the current experiments fail to evaluate the method on tasks where high-level reasoning is crucial. The existing benchmarks can be solved more effectively with other methods that do not rely on high-level reasoning.

(2) From the previous response, the authors seem to

(A) use high quality and test-related data (fractal20220817_data and bridge) for SFT cold-start and RL training according to previous response.
(B) use 100K subset with similar mixture as OXE for the action model.
What I consider bring extra advantages for the proposed method is (A) because we all know that for current embodied policy, training on test related embodiment/tasks will increase the final performance. In the authors response, the authors mainly focus on fine-tuning baselines using (B) and explain the data format for (B) which make the concern still hold.

**Quality:**

3

**Strengths And Weaknesses:**

Strengths:

- The writing is mostly clear and easy to follow.

- The evaluation especially the few-shot adaptation part seems to be interesting.

Weakness:

- The method itself seems to be a combination of two existing methods (Megma and GRPO) which limited the novelty.

- The author claims the system can bridge high level reasoning and low level actions. However, most evaluation tasks are pick-and-place and do not require serious high level reasoning.

- Some experiment settings seem to be misaligned with baselines. Need additional experiments / explanation to help clarify the source of improvements comes from the method rather than the different data mixture which might contain more test-relevant dataset. See questions below.

---

> ### Author Rebuttal · Authors · 2025-07-31
>
> ## **Response to Reviewer h8uF**
>
> **(W1) The method itself seems to be a combination of two existing methods (Magma [44] and GRPO [36]) which limited the novelty.**
>
> Thanks to the reviewer for pointing this out. For Magma, it uses trajectory prediction merely as an auxiliary training task, and it directly outputs robot actions during inference. For our work, we exploit visual trajectory latents as a long-term visual plan, that guides and enhances the action model for long-horizon robot manipulation tasks. Regarding GRPO, it is simply the RL optimization technique we adopt (commonly used across many works [11,25,39]). In other words, our innovation lies in **action-aligned visual rewards** that combine goal completion and trajectory consistency, enabling **embodied reasoning** grounded in visual-spatial dynamic scenes while unleashing capabilities like **reflection** and **self-correction** (Figure 6). A naive combination of Magma with GRPO cannot achieve reflection and self-correction due to the lack of our dual-system architecture that connects structured reasoning with executable actions through visual latent planning.
>
>
> **(W2, Q5) The main motivation of the paper is to build a system that can bridge high-level reasoning and low-level action. However, the evaluation tasks (Sipmler and LIBERO) mostly contain simple tasks like pick and place that does not require high level planning. This may hinder the proof of the final motivation.**
>
> Thanks to the reviewer for raising this concern. We would like to clarify that SimplerEnv and LIBERO do not just contain simple tasks like pick and place. For example, LIBERO-Long contains tasks such as "turn on the stove and put the frying pan on it" or "stack the right bowl on the left bowl and place them in the tray", which require high-level planning beyond primitive pick-and-place actions and multi-step sequential reasoning.
>
> To further demonstrate our advantage on complex reasoning tasks, we additionally conducted comparisons on the recently proposed RoboTwin2.0 [B] benchmark, which contains tasks that require reasoning about object affordances and tool use to complete complex manipulation scenarios (e.g., “beat block with hammer”). Using RDT [C] as our baseline action model, ThinkAct can further enhance RDT's performance on tasks requiring high-level planning, as shown in the table below:
>
> | Method     | adjust_bottle Easy | adjust_bottle Hard | beat_block_hammer Easy | beat_block_hammer Hard | lift_pot Easy | lift_pot Hard | Average |
> |------------|--------------------|---------------------|--------------------------|-------------------------|---------------|----------------|----------------|
> | RDT    | 85.6               | 77.2                | 80.4                     | 32.9                    | 81.6            | 19.5           | 62.9 |
> | **ThinkAct**   | **94.2**               | **89.6**                | **86.1**                     | **40.3**                    | **85.2**          | **22.2**           | **69.6** |
>
> [B] RoboTwin 2.0: A Scalable Data Generator and Benchmark with Strong Domain Randomization for Robust Bimanual Robotic Manipulation. arXiv 2025.
>
> [C] RDT-1B: a Diffusion Foundation Model for Bimanual Manipulation. ICLR 2025.
>
>
> **(W3, Q4) The comparison seems to not be conducted in a fair setting. ThinkAct uses a subset of OXE during SFT cold-start, RL training, and fine-tuning (100K samples). What is the detailed mixture of this subset? Are these three stages using the same data? Baselines like OpenVLA and Magma seem to be pretrained only on a wider range of OXE data and are evaluated zero-shot. To make a fair comparison, the authors should fine-tune the baselines using the same subset, which seems to contain more test-relevant data.**
>
> Thanks to the reviewer for raising this important concern about experimental fairness. We would like to clarify that the comparison setting with OpenVLA and Magma is actually fair in our current setup.
>
> To clarify our data usage across the three training stages, as detailed in Sec. B.2.1, SFT cold-start and RL training use fractal20220817_data and bridge subsets from OXE, which are selected because of their high-quality robot trajectories, and we do not require their action ground truth for these two stages. The reasoning-enhanced action adaptation stage uses 100K data points sampled from the complete OXE dataset to connect the visual latent with the action model. We note that, both OpenVLA and Magma are trained on the complete OXE dataset, which means all methods (including ours) are trained on overlapping data, ensuring a fair comparison.
>
> As suggested by the reviewer, we are happy to further conduct comparisons by fine-tuning both OpenVLA and Magma on the same 100K samples used for our reasoning-enhanced action adaptation. As shown below, ThinkAct continues to outperform both baselines even under identical fine-tuning data conditions:
>
> | Method            | Simpler-Google (VM) | Simpler-Google (VA) | Simpler-Bridge (VM) | Average |
> |---------------------|------------------------------|-----------------------------|-----------------------------|-------------|
> | OpenVLA         |         37.3                    |     43.1                      |         14.6                   | 31.7        |
> | OpenVLA (FT) |         37.7                    |     40.4                      |         16.5                   |  31.5       |
> | Magma            |         68.4                    |      62.6                     |         35.4                    | 55.5       |
> | Magma (FT)     |         70.9                   |      61.7                     |         39.2                   | 57.3        |
> | **ThinkAct**           |         **71.5**                   |       **65.1**                    |         **43.8**                   | **60.1**        |
>
>
> **(Q1) For goal completion, why would the goal reward contains a comparison on the start point?**
>
> Thanks to the reviewer for this question. While it would be straightforward to design a goal reward focusing only on the final position, we include the start point in our design to ensure that the reasoning model accurately identifies the initial state and the gripper location before planning the manipulation sequence.
>
> This design helps the model better understand both where the task begins and where it should end, encouraging the generation of visually grounded reasoning that connects initial observations with goal states. In combination with the trajectory reward, this supports the generation of coherent and executable plans that are properly anchored to the current scene configuration rather than producing abstract reasoning disconnected from the visual context.
>
>
> **(Q2) In L146-150, the authors propose to use QA-style accuracy reward. Can you provide more details about this reward?**
>
> Thanks to the reviewer for requesting these details. Following [11, 39], the QA-style accuracy reward is computed by either answer accuracy for multiple-choice QA tasks (e.g., Reflect, EgoPlan, and LLaVA-Video-178K) or averaged ROUGE-1/2/L scores for open-ended QA tasks (e.g., RoboVQA and LLaVA-Video-178K), as mentioned in Supplementary Sec. B.1.1.
>
> Once we obtain the QA reward $r_{\text{QA}}$, we use the same approach as in Eq. 3 that combines the QA-style reward with the format reward, and then optimize using GRPO. Specifically, for QA tasks, the total reward becomes: $r = 0.9r_{\text{QA}} + 0.1r_{\text{format}}$. We will add these details in the revised version to clarify the reward computation for different task types.
>
>
> **(Q3) For OXE dataset used in the cold-start phase, what are the latent embeddings $v$ for OXE dataset?**
>
> Thanks to the reviewer for this clarification request. We would like to clarify that we do not have reasoning annotations for the OXE dataset, so during the SFT cold-start phase, we directly predict trajectories without generating reasoning latent embeddings $v_t$ for OXE data.
>
> As mentioned in Sec. B.2.2, during SFT cold-start, the reasoning latent embeddings $v_t$ are only generated for datasets that contain reasoning annotations (i.e., Video-R1-CoT). For OXE trajectory data, the model learns to map visual observations and instructions directly to trajectory predictions, as the aim is to build foundational trajectory prediction capability before reinforcement learning.
>
>
> **(Q6) Misc.: It would be helpful to include the definition of $r_\text{format}$ (Eq. 3) to make the paper self-contained.**
>
> In our implementation, the format reward $r_\text{format}$ is set to 1 when the model's response adheres to the required output structure, specifically the "\<think\> … \</think\> \<answer\> … \</answer\>" format, and 0 otherwise. This follows the conventions used in DeepSeek-R1 [13] and Video-R1 [11], as also described in Supplementary Sec. B.2.2. We will include the explicit definition of $r_\text{format}$ in the revised version to make the paper self-contained.

---

> > ### Comment · Reviewer_h8uF · 2025-08-04
> >
> > Thank the authors for the response! I still have several questions remaining:
> >
> > For W2, i respectfully disagree with the authors and i still hold my opinion. In table2, the authors use 2 simulator. In SIMPLER, the authors conduct 3 tasks on google robot (open/close drawer, move near, pick coke can), and 4 tasks on bridge robot (put carrot on plate, stack blocks, put spoon on towel, put eggplant in baskect). All these tasks are pick and place tasks and can be done in 1 step. The only task in SIMPLER that requires long-horizon planning (google robot-"put the apple in closed drawer"), which requires 3 steps to finish, is otherwise omitted. Why did the authors choose to discard this task? Baselines like CogACT can be compared on this task.
> >
> > On the otherhand, the LIBERO-long tasks, which "require high-level planning beyond primitive pick-and-place actions and multi-step sequential reasoning" according to the authors. However, baselines like OpenVLA-OFT which do not use the reasoning step, is able to finish the tasks even better (+11% than ThinkACT).  Could the author compare with this method and explain the results?
> >
> > For W3, how did you select 100K high quality trajectories?

---

> ### Author Response · Authors · 2025-08-06
>
> We appreciate the reviewer's feedback, and please see our responses to the questions below.
>
> ---
>
> **(Q1) In SIMPLER, the authors conduct 3 tasks on google robot (open/close drawer, move near, pick coke can), and 4 tasks on bridge robot (put carrot on plate, stack blocks, put spoon on towel, put eggplant in baskect). Why did the only long-horizon task in SIMPLER, "put the apple in closed drawer", is omitted? Baselines like CogACT can be compared on this task.**
>
> We thank the reviewer for suggesting CogACT [E] during the discussion period. The reason why we consider the three tasks of SIMPLER is for fair comparison purposes. We followed the protocol established by the ICLR’25 work of TraceVLA [50], that uses the same subset of SimplerEnv tasks for evaluation. To further address the reviewer’s concern, we now conduct additional experiments on the 4th task of “put the apple in closed drawer” and list the performance comparisons in the following table.
>
> It can be seen that ThinkAct still outperforms SoTA methods reported in our paper, including CVPR’25 Magma [44] (e.g., ThinkAct achieved a 25.7% success rate, compared to 0.0% from the DiT-Policy baseline). This confirms that introducing additional visually aligned reasoning capability would allow action models to handle complex, sequential tasks.
>
> |                          | Octo-Base | DiT-Policy | OpenVLA | Magma | CogACT | ThinkAct |
> |--------------------------|-----------|------------|---------|--------|--------|-----------|
> | Put apple in drawer (VM) | 0.0       | 0.0        | 0.0     | 6.5    | 50.9   | 26.8      |
> | Put apple in drawer (VA) | 0.0       | 0.0        | 0.0     | 18.5   | 46.6   | 24.5      |
> | Average                  | 0.0       | 0.0        | 0.0     | 12.5   | 48.8   | 25.7      |
>
>
> Regarding the suggested CogACT [E], we note that it focuses on the action prediction model and introduces two techniques, which are orthogonal to our ThinkAct:
>
> * Action chunking: predicts action chunks instead of single-step actions, enabling smoother transitions by modeling short-term temporal dependencies and mitigating unnatural transitions between actions.
> * Adaptive action ensemble: dynamically ensembles current action predictions with past outputs using similarity-based weighting, which enhances temporal consistency and improves robustness during execution.
>
> As highlighted in Sec. 3.3, our ThinkAct aims to improve action execution by integrating the visually aligned reasoning capability from our MLLM, which produces visual latent plans (i.e., $c_t$) serving as the condition for the action model. It is worth repeating that, the above two techniques proposed in CogACT focus on improving the action model itself. These techniques are orthogonal and can be applied on top of our DiT-Policy to further boost performance. We view the two approaches as complementary rather than competing. Due to time and resource limitations during the discussion period, we have not yet been able to conduct a full integration, but we believe this is a promising direction and will consider it in future work.
>
> [E] CogACT: A Foundational Vision-Language-Action Model for Synergizing Cognition and Action in Robotic Manipulation. arXiv 2024

---

> ### Author Response · Authors · 2025-08-06
>
> **(Q2) Why do baselines like OpenVLA-OFT, which do not use a reasoning step, outperform ThinkAct on the LIBERO-long tasks, which require high-level planning beyond primitive pick-and-place actions and multi-step sequential reasoning?**
>
> We thank the reviewer for suggesting OpenVLA-OFT [F] during the discussion period. OpenVLA-OFT introduces several techniques tailored specifically for improving low-level action prediction, such as:
>
> * Parallel decoding: enables the model to predict all action tokens in a single forward pass using bidirectional attention, instead of relying on autoregressive generation.
> * Action chunking: enables the model to predict multiple consecutive actions at once, improving temporal consistency and smoothness in long-horizon tasks.
> * L1 regression loss: replaces the discrete action tokenizer with an MLP that directly regresses continuous actions using L1 loss, improving prediction stability and precision.
>
> These techniques, optimized for low-level action prediction, are again orthogonal to our ThinkAct, which enhances existing action models (i.e., DiT-Policy in our experiments) by introducing reasoning and planning capabilities. As shown in Table 1 of the original OpenVLA-OFT [F] paper, incorporating these techniques boosts LIBERO-Long performance from 53.7% to 90.7%, indicating substantial gains in action model prediction. Since these techniques are orthogonal to ThinkAct, incorporating them into our framework could potentially further enhance overall performance.
>
> It is worth noting that, however, the suggested methods of [E, F] do not incorporate explicit reasoning mechanisms and cannot “reflect” and “self-correct” during execution. In contrast, as discussed in Sec. 4.5.2, ThinkAct’s reasoning capability enables reflective decision-making and recovery from failures (see Figure 6 and our Q1 response to Reviewer K4qm regarding RoboFAC [A] experiment). As mentioned above, we view the improvements in the action model itself (as [E, F] do) would serve as complementary to our reasoning framework, and we believe integration of [E, F] and our introduced visually aligned reasoning capability would lead to stronger embodied performance.
>
> [A] RoboFAC: A Comprehensive Framework for Robotic Failure Analysis and Correction. arXiv 2025.
>
> [F] Fine-Tuning Vision-Language-Action Models: Optimizing Speed and Success. RSS 2025
>
> **(Q3) For W3, how did you select 100K high quality trajectories?**
>
> We thank the reviewer for the question and would like to further clarify the concern. For fair comparison purposes, we followed [15, 44] and utilized the same OXE training set to first pretrain our action model. During the reasoning-enhanced action adaptation stage, 100K triplets $(o_t, l, a_t)$ are **randomly sampled** from this OXE dataset to connect the visual planning latent $c_t$ produced by the reasoning MLLM with the above action model. We have verified that the distribution of the sampled subset closely matches that of the full dataset in the table below.
>
> | Subset name             | Distribution of OXE training set | Distribution of randomly selected 100K from OXE |
> |-------------------------|----------------------------------|--------------------------------------------------|
> | fractal20220817_data    | 25.6%                            | 23.7%                                            |
> | bridge_orig             | 14.5%                            | 14.9%                                            |
> | kuka                    | 13.9%                            | 14.2%                                            |
> | bc_z                    | 8.1%                             | 7.5%                                             |
> | fmb_dataset             | 7.7%                             | 7.6%                                             |
> | others (20 subsets)     | 30.2%                            | 32.1%                                            |
>
> To derive the above visual planning latent $c_t$, as discussed in Sec. B.2.1, we train the reasoning MLLM guided by 2D visual traces extracted by off-the-shelf detectors from LLARVA [27] applied to the OXE subsets of fractal20220817_data and bridge. While we refer to the above trajectories as high-quality ones in our prior response, we note that the above procedure follows the same protocol as TraceVLA [50], where no ground truth 2D visual trace is available. We will be happy to clarify this in the revised manuscript.

---

> > ### Comment · Reviewer_h8uF · 2025-08-07
> >
> > thank the authors for the response! I think the authors find an interesting way to combine high level reasoning and robot execution and the combination is not naive. However, my major concerns still hold:
> >
> > (1) the major claim is to "bridge high-level reasoning with low-level action execution via reinforced visual latent planning" However, the current experiments fail to evaluate the method on tasks where high-level reasoning is crucial. The existing benchmarks can be solved more effectively with other methods that do not rely on high-level reasoning.
> >
> > (2) From the previous response, the authors seem to
> >
> > - (A) use high quality and test-related data (fractal20220817_data and bridge) for SFT cold-start and RL training according to previous response.
> >
> > - (B) use 100K subset with similar mixture as OXE for the action model.
> >
> > What I consider bring extra advantages for the proposed method is (A) because we all know that for current embodied policy, training on test related embodiment/tasks will increase the final performance. In the authors response, the authors mainly focus on fine-tuning baselines using (B) and explain the data format for (B) which make the concern still hold.

---

> ### Author Response · Authors · 2025-08-07
>
> Thank you for the reviewer's valuable feedback. Please find our responses to the questions below.
>
> ---
>
>
> **(Q1) The major claim is to "bridge high-level reasoning with low-level action execution via reinforced visual latent planning". However, the current experiments fail to evaluate the method on tasks where high-level reasoning is crucial. The existing benchmarks can be solved more effectively with other methods that do not rely on high-level reasoning.**
>
> We understand your concern, and we would like to clarify the following points:
>
> 1. **Benchmarks used in the manuscript are for fair comparisons.**
>
>     In our manuscript, we consider SimplerEnv [19] and LIBERO [21], which are considered in recent VLA literature (e.g., ICLR’25 of TraceVLA [50], CVPR’25 of CoT-VLA [49], CVPR’25 of Magma [44]) for evaluating manipulation, including *long-horizon planning* tasks.
>
> 2. **Evaluation on additional benchmarks with more challenging tasks and requiring reasoning has been added during rebuttal.**
>
>     As suggested, we additionally consider RoboTwin2.0 [B] and RoboFAC [A], which have been recently proposed for evaluating “diverse manipulation tasks and cognitive abilities essential for robotic reasoning” [A]. For example, RoboTwin2.0 requires reasoning about *object affordances* and *tool use for complex scenarios* (e.g., reasoning about hammer affordance, identifying target regions, and planning strikes), and RoboFAC focuses on *failure detection* and *correction reasoning* (e.g., identifying failure types in planning or execution, then generating corrective plans at both high-level (sub-tasks) and low-level (actual actions)).
>
>     We would appreciate it if the reviewer could suggest benchmarks beyond the aforementioned ones, which could better demonstrate high-level reasoning capabilities.
>
> [A] RoboFAC: A Comprehensive Framework for Robotic Failure Analysis and Correction. arXiv 2025.
>
> [B] RoboTwin 2.0: A Scalable Data Generator and Benchmark with Strong Domain Randomization for Robust Bimanual Robotic Manipulation. arXiv 2025.
>
> ---
>
> **(Q2) From the previous response, the authors seem to**
> * **(A) use high quality and test-related data (fractal20220817_data and bridge) for SFT cold-start and RL training according to previous response.**
> * **(B) use 100K subset with similar mixture as OXE for the action model.**
>
> **What I consider bring extra advantages for the proposed method is (A) because we all know that for current embodied policy, training on test related embodiment/tasks will increase the final performance. In the authors response, the authors mainly focus on fine-tuning baselines using (B) and explain the data format for (B) which make the concern still hold.**
>
>
> We respectfully clarify the concern regarding the fairness of our comparisons:
>
> 1. **Same OXE training dataset for fair comparisons:**
>
>     As noted in our previous response to (W3, Q4), our training setting follows that of SOTAs like OpenVLA [15], TraceVLA [50], and Magma [44]. All methods require the **same** OXE dataset for training, and no additional action-annotated data is utilized in our work.
>
> 2. **Concerns about selecting “high-quality and test-related” training subsets.**
>
>     We understand the reviewer’s concern that, when training our MLLM for visually aligned reasoning, we followed TraceVLA [50] and considered the OXE subset of *fractal20220817_data* and *bridge* to generate 2D trajectories (as training guidance for MLLM). Although TraceVLA [50] did not view such data as “high-quality and test-related”, we agree with the reviewer that such data do share similarity with the Google Robot and Bridge environments in SimplerEnv.
>
>     It is worth pointing out that, for evaluation, we have demonstrated superior performance not only on SimplerEnv but also on benchmarks of LIBERO [21], RoboTwin2.0 [B], EgoPlan-Bench2 [6], RoboVQA [35], and OpenEQA [26]. For example, RoboTwin2.0 [B] requires complex tool manipulation, and RoboVQA [35] is for embodied reasoning. These benchmarks further address VLA and VQA problems, which are ***not*** covered in SimplerEnv. We hope this would sufficiently clarify our setting (including that of TraceVLA [50]) and address the concern of the reviewer.

---

### Official Review · Reviewer_K4qm · 2025-07-02

**Clarity:** 3
**Significance:** 3
**Originality:** 3
**Rating:** 5
**Confidence:** 5

**Summary:**

In this work, the authors present ThinkAct, a framework that reinforces visual latent planning for vision-language-action reasoning tasks. ThinkAct trains a multimodal LLM to generate embodied reasoning plans guided by reinforcing action-aligned visual rewards. Experiments show strong long-horizon planning, few-shot adaptation, and emergent behaviors like failure detection and self-correction—advancing scalable, adaptable embodied AI.

**Questions:**

**Ablations.** While the quantitative ablation study is appreciated, Table 3's ablation results for rewards (r_traj, r_goal, and SFT cold-start) show only a modest performance drop of 2-5%. This limited drop raises questions about the true importance of the reasoning MLLM in this framework.

**Table 2.**  I would like to clarify the distinction between Qwen2.5-VL* and ThinkAct. Does Qwen2.5-VL* still utilize the latent variable c_t, which is then passed to the Action Model? If not, it seems necessary to conduct an experiment comparing a high-level VLM that extracts c_t for a low-level controller (i.e., the Action Model).

**The constructed prompt closely resembles the one used in RoboPrompt.** If this is accurate, please ensure proper citation within lines 225-235.

**Reasoning Elicits Self-Correction.** To assess the importance of self-correction in robust robot manipulation, it would be beneficial to design experiments that explicitly demonstrate this capability. Such experiments would offer clearer insight into the consistency of self-corrective behavior, which is otherwise difficult to evaluate.

**Ethical Concerns:**

["NO or VERY MINOR ethics concerns only"]

**Final Justification:**

After reading the authors' feedback and other reviewers' opinions, I would like to thank the authors for their rebuttal.

My questions have been fully addressed. Therefore, I raised my score to 5, as this paper presents a potentially exciting direction for future research.

**Limitations:**

Overall, I liked this paper. However, I am worried about the importance of the GRPO Vs. a standard SFT.

**Quality:**

3

**Strengths And Weaknesses:**

1) In general, the paper is clear and the technical details are easy to understand.


2) Exploring the reinforcement of visual latent planning for downstream reasoning tasks in VLAs is significant and important.


3) The experiments section is thorough and extensive.

---

> ### Author Rebuttal · Authors · 2025-07-31
>
> ## **Response to Reviewer K4qm**
>
> **(Q1) Why do Table 3's ablation results show only a modest performance improvement of 2-5% for rewards ($r_{traj}$, $r_{goal}$) from SFT cold-start? This limited improvement raises questions about the true importance of the reasoning MLLM in this framework.**
>
> We would like to clarify that, following previous works [11,13], our reasoning framework includes both SFT cold-start and RL stages, where the SFT cold-start includes training on CoT data (i.e., Video-R1-CoT), as mentioned in Sec. B.2.2, making it already a reasoning-capable baseline. The modest improvements in Table 3 reflect the benefits of individual RL reward components ($r_{traj}$, $r_{goal}$) rather than the overall contribution of the reasoning framework itself.
>
> To properly assess the importance of our proposed reasoning framework, we additionally conduct a non-reasoning SFT baseline (Qwen2.5-VL\*\*) that is trained without any CoT data, unlike SFT cold-start, which already includes reasoning capability. The comparison below shows the true impact of our reasoning framework:
>
> | Method                   | SimplerEnv (Avg.) | LIBERO | EgoPlan-Bench2 | RoboVQA | OpenEQA | Average |
> |---------------------------|-----------------------|-------------|-----------------------|-----------------|---------------|-------------|
> | Qwen2.5-VL\*\*        | 52.8                    | 75.2        | 44.9                    | 56.4            | 49.0          | 55.7       |
> | **ThinkAct**                 | **60.1**                    | **84.4**       | **48.2**                    | **59.8**            | **56.2**          | **61.7**       |
> | $\Delta$                     | +7.3                    | +9.2        | +3.3                   | +3.4            | +7.2          | +6.0       |
>
> We note that the qualitative examples in Figure 4 demonstrate that RL performs better in complex reasoning scenarios requiring visual grounding capability. Additionally, we evaluated our ThinkAct on the RoboFAC [A] benchmark that specifically requires reasoning about failure detection and correction. This benchmark further demonstrates that our RL reasoning significantly improves upon the SFT cold-start baseline, with substantial performance gains that highlight the critical role of reinforced reasoning in embodied tasks.
>
> |Method  | Failure locating | Task identification | Failure detection | Task planning | Failure identification | Failure explanation | High-level correction | Low-level correction | Average |
> |-------------------|------------------|----------------------|-------------------|----------------|-------------------------|----------------------|------------------------|-----------------------|---------|
> | SFT cold-start        | 80.6             | 43.3                 | 79.3              | 38.7           | 35.6                    | 43.8                 | 46.4                   | 25.6                  | 49.2    |
> | **ThinkAct**      | **83.4**         | **53.9**             | **85.3**          | **48.7**       | **39.1**                | **51.3**             | **48.1**               | **28.3**              | **54.8** |
>
> [A] RoboFAC: A Comprehensive Framework for Robotic Failure Analysis and Correction. arXiv 2025.
>
>
>
> **(Q2) Does Qwen2.5-VL\* in Table 2 utilize the latent variable $c_t$ passed to the Action Model? If not, an experiment comparing a high-level VLM that extracts $c_t$ for the low-level controller seems necessary.**
>
> No, Qwen2.5-VL\* does not utilize the latent variable $c_t$ since it is a baseline that was fine-tuned on VQA datasets (i.e., EgoPlan-IT and RoboVQA) for embodied reasoning tasks in Table 2. For the evaluation of embodied reasoning benchmarks, only textual outputs for QA tasks are generated by the VLM part, with no visual latent $c_t$ and action model involved.
>
> As suggested by the reviewer, we constructed an additional baseline (denoted as Qwen2.5-VL\*\*) for this comparison. As mentioned in the response to Q1, Qwen2.5-VL\*\* is trained from Qwen2.5-VL\* on trajectory prediction datasets so that it can extract visual latent $c_t$ for a low-level controller. The comparisons below demonstrate that our action-aligned visual rewards for RL reasoning are crucial for effective latent planning:
>
> | Method                   | Simpler-Google (VM) | Simpler-Google (VA) | Simpler-Bridge (VM) | LIBERO | Average |
> |---------------------------|------------------------------|-----------------------------|-----------------------------|-------------|--------|
> | Qwen2.5-VL\*\*  | 65.3                             | 55.4                           | 37.7                          | 75.2        | 58.4 |
> | **ThinkAct**                 | **71.5**                             | **65.1**                           | **43.8**                          | **84.4**        |**66.2** |
>
> This ablation confirms that simply extracting visual latents without our action-aligned reinforcement learning achieves significantly lower performance, validating the importance of our proposed reasoning framework.
>
>
> **(Q3) The constructed prompt closely resembles the one used in RoboPrompt. If this is accurate, please ensure proper citation within L225-235.**
>
> We thank the reviewer for the suggestion. As noted in Supplementary Table 1, our prompt design is referenced from Video-R1 [11], encouraging the model to reason through the scene and generate a 2D trajectory as a visual plan. In contrast, RoboPrompt’s prompt [D] is designed to guide LLMs to directly predict robot actions through in-context learning examples without reasoning steps. We are happy to include a discussion of RoboPrompt in the Related Works section in the revised version for completeness.
>
> [D] In-context learning enables robot action prediction in LLMs. ICRA 2025.
>
>
> **(Q4) To assess the importance of self-correction in robust robot manipulation, experiments (e.g., quantitative evaluation) that explicitly demonstrate this capability would be beneficial.**
>
> Thanks to the reviewer for this suggestion. In addition to the qualitative examples shown in Figure 6 and Figure A2 demonstrating the reflection and self-correction capability, we additionally evaluated on the recently proposed RoboFAC [A] benchmark, which is specifically designed to assess failure understanding and correction in robot manipulation through a QA format. As shown in the table below, we compare our ThinkAct with the non-reasoning baseline (Qwen2.5-VL), demonstrating the effectiveness of our proposed ThinkAct for self-correction capability:
>
> |Method  | Failure locating | Task identification | Failure detection | Task planning | Failure identification | Failure explanation | High-level correction | Low-level correction | Average |
> |-------------------|------------------|----------------------|-------------------|----------------|-------------------------|----------------------|------------------------|-----------------------|---------|
> | Qwen2.5-VL        | 80.9             | 39.3                 | 83.4              | 35.6           | 36.0                    | 34.2                 | 44.7                   | 24.2                  | 47.3    |
> | **ThinkAct**      | **83.4**         | **53.9**             | **85.3**          | **48.7**       | **39.1**                | **51.3**             | **48.1**               | **28.3**              | **54.8** |
>
> The results clearly indicate that our RL-enhanced reasoning framework significantly improves failure detection and correction abilities compared to standard VLMs, providing quantitative evidence for the self-correction capabilities observed qualitatively in our manipulation experiments (Figure 6 and Figure A2).

---

> > ### Comment · Reviewer_K4qm · 2025-08-01
> >
> > After reading the authors' feedback and other reviewers' opinions, I would like to thank the authors for their rebuttal.
> >
> > My questions have been fully addressed. Therefore, I raised my score to 5, as this paper presents a potentially exciting direction for future research.

---

> ### Author Response · Authors · 2025-08-01
>
> Thank you for your valuable review and for taking the time to read our responses.
>
> We appreciate that our rebuttal addressed your concerns and will incorporate the clarifications and suggested references into our revised version.
>
> Thank you again for your constructive feedback.

---

### Official Review · Reviewer_MRi2 · 2025-07-12

**Clarity:** 3
**Significance:** 3
**Originality:** 3
**Rating:** 4
**Confidence:** 2

**Summary:**

The paper introduces ThinkAct, a dual-system framework that integrates high-level reasoning with low-level action execution for vision-language-action (VLA) tasks. The key innovation lies in using reinforcement learning with action-aligned visual rewards to train a multimodal LLM to generate visual plan latents, which then guide a downstream action model. The approach is evaluated on simulated robot manipulation and embodied reasoning benchmarks, demonstrating improvements in long-horizon planning, few-shot adaptation, and self-correction.

**Questions:**

See Strengths And Weaknesses

**Ethical Concerns:**

["NO or VERY MINOR ethics concerns only"]

**Final Justification:**

The authors have addressed my concerns, I keep my rating.

**Limitations:**

Yes

**Quality:**

3

**Strengths And Weaknesses:**

Strengths

- The separation of reasoning (via LLM) and action execution (via policy) is well-motivated and aligns with cognitive science principles of "slow thinking" and "fast control."

- The use of goal completion and trajectory consistency as reward signals is intuitive and empirically effective, particularly for manipulation tasks.

- It outperforms prior methods (e.g., OpenVLA, CoT-VLA) on LIBERO and SimplerEnv, with notable gains in long-horizon tasks (e.g., LIBERO-Long).

Weaknesses

- The reward function (Eq. 1–3) relies on Euclidean distance and DTW metrics, which may not generalize to non-kinematic or abstract tasks (e.g., "open the door" requires contact-rich dynamics beyond 2D trajectory matching). The paper does not discuss extending rewards to such scenarios.

- All experiments are conducted in simulation (SimplerEnv, LIBERO). While these benchmarks are standard, the lack of real-robot deployment raises concerns about sim-to-real gaps (e.g., visual discrepancies, control latency).

- Results are averaged over only 3 seeds without standard deviations or confidence intervals (Table 1). This is insufficient for noisy robotics benchmarks, especially with high-variance tasks like "Stack Blocks" (8.7% success).

---

> ### Author Rebuttal · Authors · 2025-07-31
>
> ## **Response to Reviewer MRi2**
>
> **(W1) The reward function (Eq. 1–3) relies on Euclidean distance and DTW metrics, which may not generalize to non-kinematic or abstract tasks (e.g., "open the door" requires contact-rich dynamics beyond 2D trajectory matching).**
>
> Thanks to the reviewer for pointing this out. We agree that incorporating contact-rich signals (e.g., depth information or affordance maps) would be beneficial for tasks like "open the door" that require precise interaction dynamics beyond 2D trajectory matching.
> It is worth noting, however, that we observe the reflection and self-correction capabilities learned from our reinforcement reasoning effectively alleviate such challenges. As shown in Figure 6 and A2, when objects cannot be grasped or are dropped during execution, ThinkAct detects the failure and replans to achieve success, resulting in SoTA performance on robot manipulation benchmarks, SimplerEnv (60.1%) and LIBERO (84.4%), which already include various contact-rich tasks.
>
> Additionally, our reward framework can be readily extended to incorporate contact-rich signals into the total reward function (Eq. 3). The proposed action-aligned visual reward allows extension with additional reward components that capture contact-rich dynamics. We will include a discussion of incorporating contact-rich rewards for complex manipulation tasks in our revised version, and leave it for future research.
>
>
> **(W2) While the evaluation benchmarks are standard (i.e., SimplerEnv, LIBERO), the lack of real-robot deployment raises concerns about sim-to-real gaps (e.g., visual discrepancies, control latency).**
>
> Thanks to the reviewer for raising this concern. As noted in [19], SimplerEnv has been shown to have performance highly correlated with real-robot deployment. We achieve state-of-the-art performance on these benchmarks across various visual variances (in Table 1) with only slightly increased inference latency compared to OpenVLA, as detailed in Sec. C.7 in the supplementary material.
>
> In addition to simulation benchmarks, we have included comparisons on RoboVQA in Table 2, which is actually a real-robot benchmark for long-horizon reasoning tasks, where ThinkAct demonstrates superior performance with 59.8 BLEU score compared to existing methods.
>
> To further demonstrate the real-world effectiveness of ThinkAct, we additionally conducted experiments on recently proposed benchmarks, including RoboFAC [A], a comprehensive benchmark for failure detection and correction on real-robot deployment, and RoboTwin2.0 [B], which specifically features its sim-to-real transfer characteristics for robot manipulation tasks. For the RoboTwin2.0 evaluation, we adopt RDT [C] as our baseline action model and demonstrate how ThinkAct enhances RDT's performance through our reasoning framework. As shown in the quantitative comparisons below, ThinkAct consistently outperforms existing methods, validating its potential for real-world deployment.
>
> **RoboFAC**
> |Method  | Failure locating | Task identification | Failure detection | Task planning | Failure identification | Failure explanation | High-level correction | Low-level correction | Average |
> |-------------------|------------------|----------------------|-------------------|----------------|-------------------------|----------------------|------------------------|-----------------------|---------|
> | Qwen2.5-VL        | 80.9             | 39.3                 | 83.4              | 35.6           | 36.0                    | 34.2                 | 44.7                   | 24.2                  | 47.3    |
> | **ThinkAct**      | **83.4**         | **53.9**             | **85.3**          | **48.7**       | **39.1**                | **51.3**             | **48.1**               | **28.3**              | **54.8** |
>
> **RoboTwin2.0**
> | Method     | adjust_bottle Easy | adjust_bottle Hard | beat_block_hammer Easy | beat_block_hammer Hard | lift_pot Easy | lift_pot Hard | Average |
> |------------|--------------------|---------------------|--------------------------|-------------------------|---------------|----------------|----------------|
> | RDT    | 85.6               | 77.2                | 80.4                     | 32.9                    | 81.6            | 19.5           | 62.9 |
> | **ThinkAct**   | **94.2**               | **89.6**                | **86.1**                     | **40.3**                    | **85.2**          | **22.2**           | **69.6** |
>
> [A] RoboFAC: A Comprehensive Framework for Robotic Failure Analysis and Correction. arXiv 2025.
>
> [B] RoboTwin 2.0: A Scalable Data Generator and Benchmark with Strong Domain Randomization for Robust Bimanual Robotic Manipulation. arXiv 2025.
>
> [C] RDT-1B: a Diffusion Foundation Model for Bimanual Manipulation. ICLR 2025.
>
>
>
> **(W3) Results are averaged over only 3 seeds without standard deviations or confidence intervals (Table 1). This is insufficient for noisy robotics benchmarks, especially with high-variance tasks like "Stack Blocks" (8.7% success).**
>
> Thanks to the reviewer for this concern. As mentioned in L604, our evaluation protocol follows standard practices from OpenVLA and TraceVLA that report accuracy averaged over 3 seeds.
>
> We would like to clarify that for tasks like "Stack Blocks" in SimplerEnv, each task already contains 24 different initial configurations, so the reported scores are actually averaged over 72 trials (3 seeds × 24 configurations). Similarly, each task in LIBERO contains 1500 evaluation trials (see L604), providing substantial statistical sampling. We will include the aforementioned details in our revision.
>
> To further verify the statistical significance of our evaluation setup, we conducted additional experiments with 5 seeds and report the standard deviations in the table below. The means and standard deviations between the 3-seed and 5-seed setups are similar, confirming that our evaluation protocol provides reliable results:
>
> | Method                   | Simpler-Google (VM) | Simpler-Google (VA) | Simpler-Bridge (VM) | LIBERO     |
> |---------------------------|------------------------------|----------------------------|-----------------------------|-----------------|
> | OpenVLA               | 37.3                             | 43.1                          | 14.6                          | 76.5            |
> | Magma                   | 68.4                             | 62.6                         | 35.4                           | -                  |
> | ThinkAct (3 seeds) | 71.5 ± 2.6                   | 65.1 ± 3.4                 | 43.8 ± 5.0                  | 84.4 ± 1.7   |
> | ThinkAct (5 seeds) | 71.4 ± 1.9                   | 65.3 ± 2.5                 | 43.6 ± 3.7                  | 84.1 ± 1.9   |
>
> This demonstrates that our reported improvements are statistically robust and not due to random variance.

---

### Note · Authors · 2025-08-13

We thank the reviewers and AC for their constructive feedback. The rebuttal period has been exceptionally productive, allowing us to clarify critical aspects and the contributions of ThinkAct.

---

**Recognized Strengths**

1. **Motivation is well-motivated, significant, and meaningful** `MRi2`, `K4qm`, `nnRq`.
2. **Intuitive and effective method** `MRi2`, `nnRq`.
3. **Comprehensive experiments with notable improvements** `K4qm`, `h8uF`.
4. **Clear presentation** `K4qm`, `h8uF`.

---

**Clarifications to Reviewer Concerns**

1. **Reward Generalization.** Our trajectory-based rewards already handle contact-rich tasks (SimplerEnv 60.1%, LIBERO 84.4%) via learned self-correction. It can be readily extended with contact-rich signals for more complex tasks, which we will discuss in the revised version. `MRi2` provided the acknowledgement.
2. **Improvement over SFT Cold Start.** SFT cold-start is already reasoning-capable due to CoT training, so Table 3 reflects only individual reward components. Against a true non-reasoning baseline, ThinkAct shows substantial gains across benchmarks. `K4qm`, `nnRq` acknowledged this clarification.
3. **Quantitative Self-Correction.** On RoboFAC [A], ThinkAct outperforms the baseline by +7.5%, quantifying improved self-correction. `K4qm` recognized the evaluation.
4. **Novelty** Beyond visual trajectories and GRPO, our novelty is using visual plan latents for long-term guidance with action-aligned visual rewards, enabling embodied reasoning, reflection, and self-correction capabilities. `h8uF` provided the acknowledgement.

---

**Discussion-Period Follow-ups**

* **Evaluation on Simple Tasks.** SimplerEnv and LIBERO are standard manipulation benchmarks including long-horizon tasks. We added RoboTwin2.0 [B] and RoboFAC [A], which require affordance reasoning, tool use, and corrective planning, and observed consistent gains. `h8uF` provided the acknowledgement.
* **“High-Quality, Test-Related” Training Data.** We used the same OXE dataset as prior VLA works and followed ICLR’25 TraceVLA’s OXE subset for 2D visual traces. While we agree that such training subset exhibits similarity with the Google Robot and Bridge environments in SimplerEnv, ThinkAct’s improvements extend beyond SimplerEnv to diverse VLA and VQA benchmarks, alleviating fairness and generalization concerns. `h8uF` provided the acknowledgement.

We will integrate these points and new results into the revision. Thank AC and reviewers again for the careful reviews.

---

### Decision · Program_Chairs · 2025-09-17

**Decision:**

Accept (poster)

**Comment:**

The paper proposes a framework that links reasoning and action execution via reinforced visual latent planning for embodied AI. The main strengths highlighted across reviews are the strong motivation, clear presentation, and extensive experiments demonstrating improved long-horizon planning, self-correction, and few-shot adaptation. Reviewers agreed that the idea of separating reasoning (via LLM) and execution (via action model) is well motivated and that the use of action-aligned visual rewards is intuitive and effective. A recurring weakness, however, was that the evaluation did not always convincingly demonstrate high-level reasoning beyond pick-and-place tasks, and some reviewers were concerned about fairness in the experimental setup and the reliance on potentially test-related data. One reviewer also emphasized that the novelty could be overstated, given overlaps with prior methods like Magma and GRPO, and questioned whether the improvements mainly came from architectural design rather than the RL step.

During rebuttal and discussion, the authors provided substantial clarifications and new results. They extended evaluation to harder tasks, which involve tool use, affordance reasoning, and failure correction, showing consistent gains. They also clarified the fairness of their data usage and added comparisons where baselines were fine-tuned on the same subsets, with ThinkAct still outperforming them. Reviewers who were initially skeptical (e.g., about the role of RL) acknowledged that the new experiments showed clear benefits beyond SFT cold-start, and one raised their score after rebuttal. While one reviewer’s concern about fully validating high-level reasoning remained partly unresolved, the majority view was that the authors sufficiently addressed the main criticisms and the contribution is valuable to the embodied AI community. Overall, given the well-motivated method, thorough experimental clarifications, and demonstrated improvements across multiple benchmarks, I recommend acceptance.